# Structure of formylpeptide receptor 2-G_i complex reveals insights into ligand recognition and signaling

Youwen Zhuang[1,2,3,9], Heng Liu[4,9], X. Edward Zhou[3], Ravi Kumar Verma[5], Parker W. de Waal[3], Wonjo Jang [6], Ting-Hai Xu [3], Lei Wang [4], Xing Meng[7], Gongpu Zhao [7], Yanyong Kang[3,8], Karsten Melcher [3], Hao Fan[5], Nevin A. Lambert [6], H. Eric Xu[1,3✉] & Cheng Zhang [4✉]

Formylpeptide receptors (FPRs) as G protein-coupled receptors (GPCRs) can recognize formylpeptides derived from pathogens or host cells to function in host defense and cell clearance. In addition, FPRs, especially FPR2, can also recognize other ligands with a large chemical diversity generated at different stages of inflammation to either promote or resolve inflammation in order to maintain a balanced inflammatory response. The mechanism underlying promiscuous ligand recognition and activation of FPRs is not clear. Here we report a cryo-EM structure of FPR2-G_i signaling complex with a peptide agonist. The structure reveals a widely open extracellular region with an amphiphilic environment for ligand binding. Together with computational docking and simulation, the structure suggests a molecular basis for the recognition of formylpeptides and a potential mechanism of receptor activation, and reveals conserved and divergent features in G_i coupling. Our results provide a basis for understanding the molecular mechanism of the functional promiscuity of FPRs.

[1] The CAS Key Laboratory of Receptor Research, Shanghai Institute of Materia Medica, Chinese Academy of Sciences, Shanghai 201203, China. [2] University of Chinese Academy of Sciences, Beijing 100049, China. [3] Center for Cancer and Cell Biology, Program for Structural Biology, Van Andel Research Institute, Grand Rapids, MI 49503, USA. [4] Laboratory for GPCR Biology, Department of Pharmacology and Chemical Biology, University of Pittsburgh School of Medicine, University of Pittsburgh, Pittsburgh, PA 15261, USA. [5] Bioinformatics Institute (BII), Agency for Science, Technology and Research (A*STAR), Singapore, Singapore. [6] Department of Pharmacology and Toxicology, Medical College of Georgia, Augusta University, Augusta, GA 30912, USA. [7] David Van Andel Advanced Cryo-Electron Microscopy Suite, Van Andel Research Institute, Grand Rapids, MI 49503, USA. [8] Present address: Takeda Research, 9625 Towne Centre Drive, San Diego, CA 92130, USA. [9] These authors contributed equally: Youwen Zhuang, Heng Liu. ✉email: Eric.Xu@simm.ac.cn; chengzh@pitt.edu

Formylpeptides with N-terminal formylated (N-formyl) methionine are abundantly present in bacterial or host mitochondrial proteins. They function as the major chemotactic pathogen- and damage-associated molecular patterns that can be recognized by the family of formylpeptide receptors (FPRs)[1,2]. The FPRs belong to a subfamily of G protein-coupled receptors (GPCRs), which comprises three members, FPR1, FPR2, and FPR3. FPR1 and FPR2 were initially discovered and characterized to recognize various formylpeptides to play important roles in host defense and clearance of damaged host cells, while the function of FPR3 is largely unknown[2]. Numerous studies have shown that FPR1 and FPR2 can also recognize other endogenous ligands besides formylpeptides to regulate many functional aspects of immune cells of the myeloid lineage and play multiple roles in inflammation[2–5]. FPRs together with receptors for the complement C5a peptide (C5aR), the eicosanoid lipid molecules leukotriene B$_4$ and prostaglandin D$_2$ (BLTs and CRTH2), and chemokine molecules (chemokine receptors) constitute a group of G$_i$-coupled chemoattractant receptors that belong to the γ-subgroup of rhodopsin-like Class A GPCRs[6]. Although structures of several receptors in this group have been solved, no structure of signaling complex has been reported for this group of receptors.

Among all GPCRs, FPR2 is remarkably versatile and promiscuous[3,5,7]. It can recognize diverse formylpeptides derived from various bacteria and hosts such as phenol-soluble modulins (PSMs) from highly pathogenetic *Staphylococcus aureus*[8] and mitocryptide-2 (MCT-2) from host mitochondria[9]. Beside pattern recognition, FPR2 can also recognize a variety of structurally and functionally distinct non-formylated peptides from viruses including human immunodeficiency virus (HIV)[10] and hosts. In addition, FPR2 has been shown by numerous experiments to be the receptor for bioactive eicosanoid lipid molecules such as lipoxin A$_4$ (LXA$_4$) and resolvin D$_1$ (RvD$_1$), known as the specialized pro-resolving lipid mediators (SPMs)[11], although contradictory experimental data has been reported[12,13]. FPR2 is also referred as FPR2/ALX, in which ALX means the receptor for LXA$_4$[1,14]. While most peptide ligands such as formylpeptides act on FPR2 to induce chemotaxis of immune cells and initiate numerous inflammatory processes, FPR2 signaling by SPMs promotes the resolution of inflammation[2,3,15]. A host-derived lipid-binding protein involved in the anti-inflammatory action of glucocorticoids, annexin A1, and its derived peptides, can also act on FPR2 as pro-resolving or anti-inflammatory agents[16]. It has been speculated that different endogenous ligands act on FPR2 to induce distinct signaling pathways to either promote or resolve inflammation. Because of such complex functional roles, FPR2 has been linked to many inflammation-related diseases, including asthma[17], influenza[18], Alzheimer's disease[19], and various cardiovascular diseases[20]. Therefore, there have been intensive research efforts in developing synthetic FPR2 ligands as drugs[7]. In particular, biased FPR2 agonists that can specifically activate the resolution pathways represent a new therapeutic frontier[21].

The molecular mechanisms underlying promiscuous ligand recognition and multifaceted signaling of FPRs are largely unclear due to a lack of structural understanding of the function of FPRs. Here, we report a cryo-EM structure of human FPR2-G$_i$ signaling complex with a synthetic pro-inflammatory peptide agonist. The structure together with biochemical studies and computational docking results reveal how FPR2 recognizes peptide ligands. The results also provide structural insights into receptor activation and G$_i$ protein coupling for the FPR family.

## Results

**Cryo-EM structure determination of FPR2-G$_i$ complex.** The coupling of FPR2 to G$_i$ protein induced by various agonists was characterized based on cell-based functional assays using pertussis toxin[22–25]. We performed GTPγS binding assays using cell membranes expressing human FPR2 and purified G$_i$ protein to prove that a synthetic peptide agonist with the sequence Trp-Lys-Tyr-Met-Val-d-Met-NH$_2$ (WKYMVm)[26–28] can promote the activation of Gi through FPR2 in a dose-dependent manner (Fig. 1a). As a control, it didn't induce G$_i$ activation with another G$_i$-coupled chemoattractant receptor, the C5aR (Fig. 1a), suggesting that the effects of WKYMVm-induced Gi activation was through FPR2.

We then assembled the complex of human FPR2 and heterotrimeric G$_i$ with WKYMVm for our structural studies. The complex was formed on the membrane of insect cells, treated with apyrase to hydrolyze nucleotides and purified in detergent buffers (Supplementary Fig. 1A and B). We used human Gαi1 with two dominant negative mutations[29], rat Gβ1 and bovine Gγ2 to form the Gi heterotrimer. We added an antibody fragment scFv16 to stabilize the nucleotide-free complex by binding to the interface between Gα$_i$ and Gβ[30] (Supplementary Fig. 1B). We then determined the structure of FPR2-G$_i$-scFv16 with a global nominal resolution of 3.17 Å (Table 1 and Supplementary Fig. 2). The clear density maps allowed us to unambiguously model the peptide ligand WKYMVm and most residues of FPR2 from G21 to L317 including all three

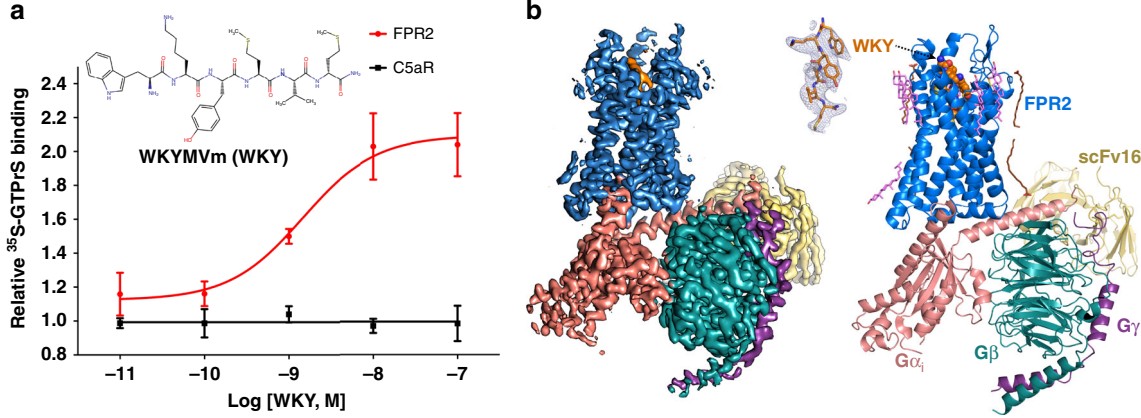

**Fig. 1 FPR2-Gi coupling and overall structure of the complex. a** $^{35}$S-GTPγS binding assays using FPR2- and C5aR-expressing cell membranes and purified G$_i$ heterotrimer with increasing concentrations of WKYMVm (WKY for short in all figures). $n = 3$, data are mean ± s.e.m. Source data are provided as a Source Data file. **b** 3D cryo-EM density map and overall structure of FPR2-G$_i$ with WKYMVm and scFv16. Cholesterol molecules are shown as pink sticks. Cryo-EM map is colored according to different subunits. The density of WKYMVm is shown in the upper middle.

**Table 1 Cryo-EM data collection, model refinement and validation statistics.**

| | FPR2-Gi1-scFV16-WKY (EMDB- EMD-20126) (PDB 6OMM) |
|---|---|
| Data collection and processing | |
| Magnification | 130,000 |
| Voltage (kV) | 300 |
| Electron exposure (e−/Å$^2$) | ~67 |
| Defocus range (μm) | 0.6-2.5 |
| Pixel size (Å) | 1.029 |
| Symmetry imposed | None |
| Initial particle images (no.) | 1,231,594 |
| Final particle images (no.) | 203,133 |
| Map resolution (Å) | 3.17 |
| FSC threshold | 0.143 |
| Map resolution range (Å) | 50-3.2 |
| Refinement | |
| Initial model used (PDB code) | 6DDE |
| Model resolution (Å) | 3.17 |
| FSC threshold | 0.143 |
| Model resolution range (Å) | 50-3.2 |
| Map sharpening *B* factor (Å$^2$) | -117.3 |
| Model composition | |
| Non-hydrogen atoms | 9134 |
| Protein residues | 1142 |
| Ligands | 8 |
| *B* factors (Å$^2$) | |
| Protein | 74.7 |
| Ligand | 74.0 |
| R.m.s. deviations | |
| Bond lengths (Å) | 0.006 |
| Bond angles (°) | 1.153 |
| Validation | |
| MolProbity score | 1.48 |
| Clashscore | 4.74 |
| Poor rotamers (%) | 0.0 |
| Ramachandran plot | |
| Favored (%) | 96.44 |
| Allowed (%) | 3.56 |
| Disallowed (%) | 0.0 |

extracellular loops (ECLs) and three intracellular loops (ICLs), the whole G$_i$ heterotrimer except for the α-helical domain (AHD) in Gα$_i$ in the structure (Supplementary Fig. 3 and Fig. 1b). The scFv16 binds to the same position on G$_i$ that is far away from the receptor and G protein interface as observed in previous structures[31,32]. No clear density is observed for the first 20 FPR2 residues, indicating a disordered N-terminus, which is similar to it observed in the structures of two closely related receptors, BLT1 and C5aR[33,34].

**A widely open ligand-binding pocket for WKYMVm.** WKYMVm is among the most potent peptide agonists of FPR2 characterized so far[1]. It has shown positive therapeutic effects in tissue repair and regeneration in a number of animal-based models[35]. In the structure, WKYMVm adopts an extended conformation to bind in a heart-shaped ligand-binding pocket with the top region widely open to the extracellular milieu (Fig. 2a). The bulky side chains of the tryptophan and tyrosine residues and the long and extended side chains of the lysine and methionine residues at the N-terminal segment of WKYMVm occupy the top region of the ligand-binding pocket, while the C-terminal valine and d-methionine residues insert into the narrow bottom region of the ligand-binding pocket (Fig. 2b, c). Especially, compared to other chemoattractant GPCRs bound to peptide ligands, the C-terminal end of WKYMVm inserts more deeply into the receptor

core (Fig. 2d), which allows it to directly contact the conserved residue W254$^{6.48}$ (Fig. 2c). Every residue in WKYMVm is involved in direct hydrophobic or polar interactions with surrounding residues of the receptor, allowing the peptide agonist to make contract with ECLs 1 and 2 and TM3-7. Such extensive interactions may account for the extremely high affinity of WKYMVm for FPR2[24,27]. Another synthetic peptide WKYMVM with an L-methionine instead of D-methionine at the C-terminus has been shown to bind to FPR2 with an about 20-fold lower affinity[26]. In our structure, the L-methionine would put the acetamide group of the ligand away from D106$^{3.33}$ (Ballesteros–Weinstein numbering[36]) of FPR2 and thus disrupt the hydrogen bond, resulting in a lower affinity.

The WKYMVm binding pocket exhibits an amphiphilic environment. There are mainly two clusters of hydrophobic residues from the top to the bottom of the ligand-binding pocket that direct contact WKYMVm. The first cluster consists of residues F178, L198$^{5.35}$, L268, and L272 from ECL2, ECL3 and TM5 (Fig. 2b). They form a hydrophobic interaction network with the side chain of Trp residue in WKYMVm on the top of the ligand-binding pocket. The second cluster is larger, consisting of residues F257$^{6.51}$ and V284$^{7.35}$ on the side of ligand-binding pocket and residues L109$^{3.36}$, F110$^{3.37}$, W254$^{6.48}$, and F292$^{7.43}$ at the bottom to interact with the side chains of Val and D-Met residues of WKYMVm (Fig. 2c). The side chain of Tyr residue in WKYMVm forms π-stacking interactions with H102$^{3.92}$ and F178 of FPR2. The acetamide group of the last D-Met residue of WKYMVm forms an extensive polar interaction network with D106$^{3.33}$, R201$^{5.38}$, and R205$^{5.42}$ of FPR2 (Fig. 2c). Our results agree with the results from previous computational modeling and docking studies[37,38]. To further validate ligand-binding mode, we measured WKYMVm-induced activation of FPR2 with mutations of residues involved the interactions with WKYMVm. Some of the mutations led to little expression of the receptor on HEK-293 cell surface; others affected the WKYMVm-induced receptor activation to various extents, consistent with our structural findings (Supplementary Fig. 4A and B). It is to be noted that the mutations of two arginine residues, R201$^{5.38}$ and R205$^{5.42}$, which form polar interactions with the acetamide group of D-Met residue of WKYMVm, significantly compromised WKYMVm-induced FPR2 activation (Supplementary Fig. 4C). These two residues may be involved in the recognition of formyl group of formylpeptides, which will be discussed in the next section.

To investigate the conformational dynamics of FPR2 in ligand binding, we performed molecular dynamics (MD) simulations on FPR2 alone without ligand and G$_i$ protein at a timescale of 2 μs in duplicate. The overall structure of FPR2 underwent significant conformational changes at different scales in two simulations (Supplementary Fig. 5A). In both simulations, the cytoplasmic end of TM6 moved towards TM2 (Supplementary Fig. 5B), which suggests a relaxation of TM6 towards the inactive conformational state likely due to the removal of agonist and G$_i$ protein[39]. The extracellular region showed large structural fluctuations with ECL3 and ECL2 regions moving closer to each other in both simulations (Supplementary Fig. 5C). It is difficult to speculate on whether these results represent real conformational states sampled by unliganded FPR2 because of the short timescale of simulations, but they imply a highly flexible nature of the extracellular region of FPR2, which may allow the receptor to recognize chemically diverse agonists in different ways.

**Recognition of formylpeptides by FPRs.** To further investigate how FPRs recognize formylpeptides, we performed computational docking studies to dock six short formylpeptides, fMLF, fMLFII, fMLFIK, fMLFK, fMLFW, and fMLFE, into the structure

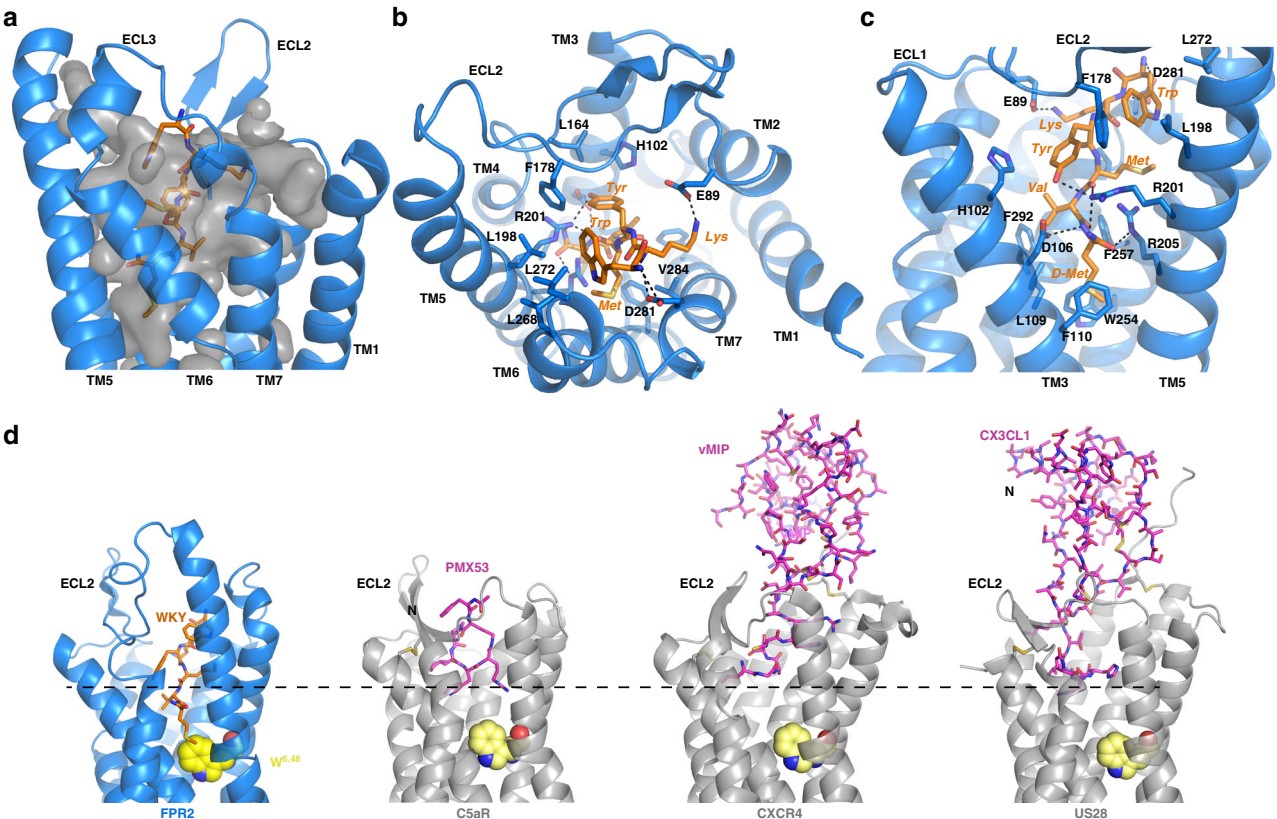

**Fig. 2 Binding pocket for WKYMVm. a** The overall shape of binding pocket for WKYMVm is shown in gray. **b** Binding pose of WKYMVm viewed from the extracellular surface. **c** Binding pose of WKYMVm viewed from the side of 7-TM bundle. Residues in FPR2 are labeled with 1-letter names followed by sequence numbers; residues in WKYMVm are labeled with 3-letter names in italics. Polar and hydrogen bonding interactions are shown as black dashed lines. **d** Structural comparison of chemoattractant GPCRs with peptide ligands. From left to right: FPR2 with agonist WKYMVm (WKY), C5a receptor (C5aR) with antagonist PMX53 (PDB ID 6C1R), chemokine receptor CXCR4 with chemokine antagonist vMIP (PDB ID 4RWS), viral chemokine receptor US28 with chemokine agonist CX3CL1 (PDB ID 4XT1). The conserved W[6.48] residue in each receptor is shown as yellow spheres.

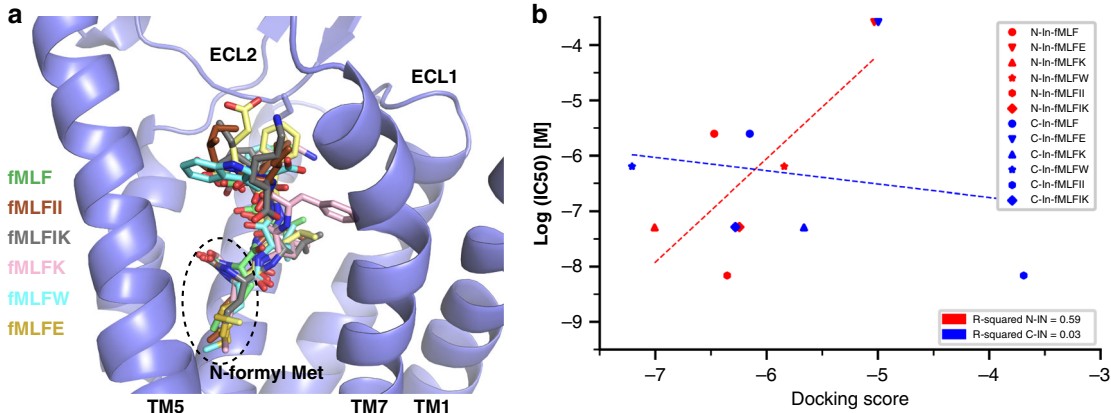

**Fig. 3 Docking of formylpeptides to FPR2. a** Binding poses of six formylpeptides in the N-ter-in mode in FPR2 (slate). The N-terminal formyl (N-formyl) Met residues are circled. **b** The computed glide docking scores for six formylpeptides in the N-ter-in (red) and C-ter-in (blue) modes are plotted against the Log ($IC_{50}$) values of them in the cAMP accumulation assays reported previously[40].

of FPR2 (Fig. 3a). The pharmacological action of these peptides on FPR2 has been defined previously through multiple methods including the cAMP accumulation assays[40]. Initially, we sought to determine whether the formylpeptides bind to FPR2 with the C-terminus-inside (C-ter-in) mode as WKYMVm, or with the N-terminus-inside (N-ter-in) mode. Although the latter is associated with an inverted orientation compared to WKYMVm, it would place the N-formyl methionine residue of formylpeptides at the

similar position as the D-methionine residue of WKYMVm (Fig. 3a). We docked the six formylpeptides to FPR2 with both modes and calculated the glide docking scores[41], which were then compared to the $IC_{50}$ values of these peptides in inhibiting cAMP accumulation[40] (Supplementary Table 1). Our results show that the log($IC_{50}$) values for the formylpeptides correlate well with the docking scores obtained for the N-ter-in mode but not for the C-ter-in mode (Fig. 3b), indicating that the formylpeptides prefer to

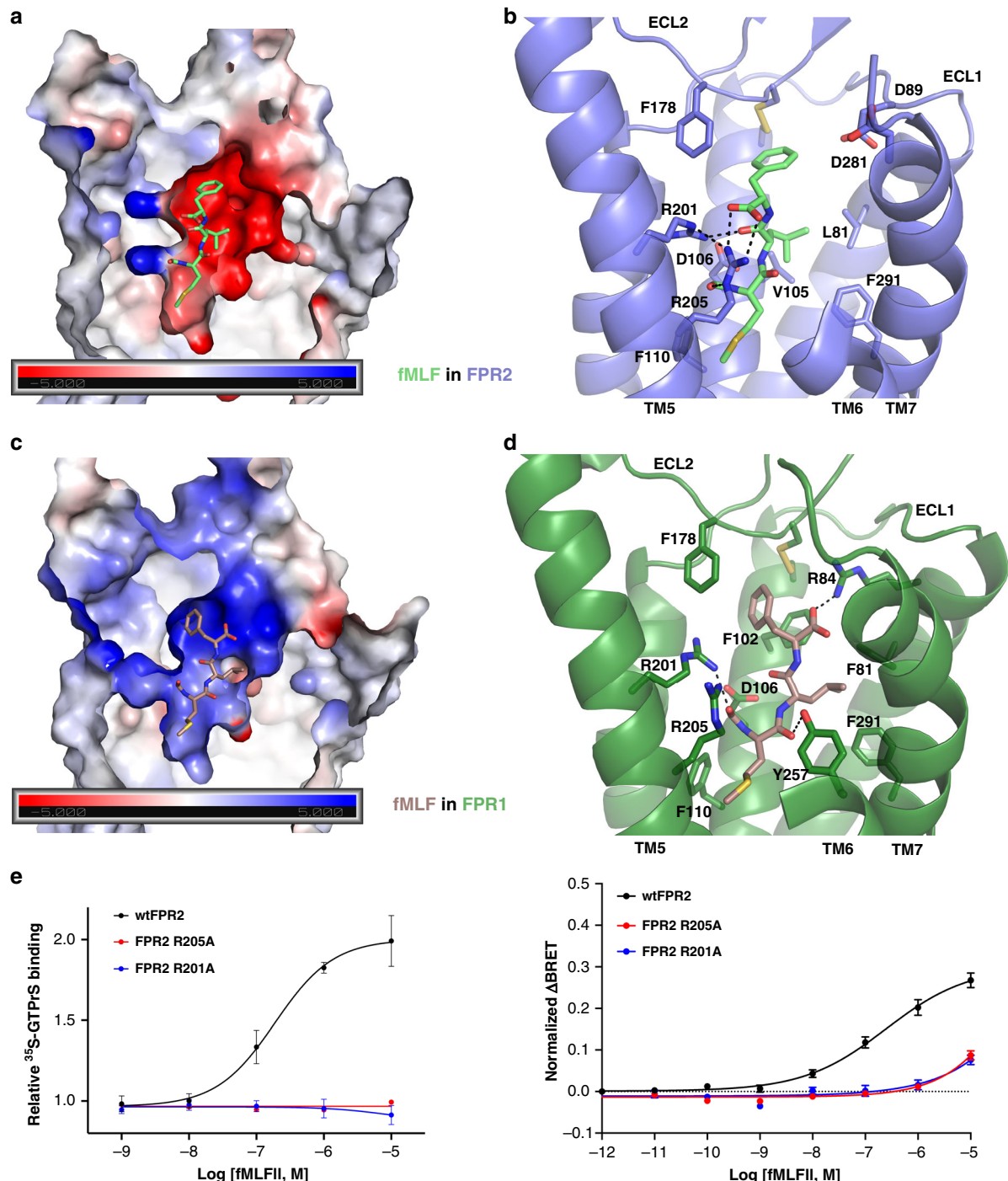

**Fig. 4 Peptide-binding pockets in FPR1 and FPR2. a** Negatively charged environment of the ligand-binding pocket in FPR2 and docked fMLF peptide. **b** Molecular details of the ligand-binding pocket in FPR2 for fMLF. **c** Positively charged environment of the ligand-binding pocket in FPR1 and docked fMLF peptide. **d** Molecular details of the ligand-binding pocket in FPR1 for fMLF. Hydrogen-bonding interactions are shown as dashed lines. **e** Effects of mutations R205A and R201A on fMLFII-induced receptor activation determined by $^{35}$S-GTPγS binding assays (left) and BRET-based assays (right). BRET-based assays measured $G_i$ coupling to receptors. $n = 3$–10, data are mean ± s.e.m. Source data are provided as a Source Data file.

bind FPR2 with the N-ter-in mode. In addition, results from previous docking and simulation studies as well as mutagenesis studies predicted the same N-ter-in binding mode of for-mylpeptides, consistent with our studies[37,38,40].

Such a binding mode well explains the selectivity of FPR2 for different formylpeptides. It has been established that in general FPR2 prefers long formylpeptides over short formylpeptides[1,3,42]. The docking results suggest that for short formylpeptides such as fMLF, the prototypical formyl peptide used in functional studies

of FPRs[1], their C-terminal carboxyl groups would be located in a negatively charged environment attributed to residues E89 and D281[7.32] at the top region of the ligand-binding pocket of FPR2 (Fig. 4a, b), which is energetically unfavorable. In our docking structure of fMLF, its C-terminal carboxyl group forms hydrogen bonds with R205[5.42], placing the side chain of Phe in fMLF towards E89 and D281[7.32] (Fig. 4b). The negatively charged environment of the ligand-binding pocket also leads to the selectivity of FPR2 for short formylpeptides with positively

charged residues at the C-terminus, e.g. the potency of fMLFK in activating FPR2 is about 500–5000-fold higher than that of fMLFE[40]. Consistently, a mutation of D281[7.32]G has been shown to greatly increase the affinities of fMLF and fMLFE for FPR2[40] (Supplementary Table 1). Among all six formylpeptides used in our docking studies, fMLFII and fMLFIK showed the highest potencies in activating FPR2 (Supplementary Table 1). The docking results suggest that the two C-terminal hydrophobic residues in fMLFII form extensive hydrophobic interactions with hydrophobic residues in the ligand-binding pocket of FPR2, and the last Lys residue in fMLFIK forms a hydrogen bond with D281[7.32], which may lead to their higher affinities for FPR2 compared to fMLF[40] (Supplementary Fig. 6). D281[7.32]G mutation has been shown to greatly decrease the affinities of fMLFIK and fMLFK for FPR2[40] (Supplementary Table 1), supporting our docking results.

FPR1 and FPR2 share a high 69% sequence identity, but they have distinct preferences for formylpeptides and play different roles in host defense[4,40,43]. The extracellular region of FPR2 including all extracellular termini of 7-TMs and ECLs exhibits a much larger sequence diversity compared to the 7TM core region and the cytoplasmic when aligned with FPR1, providing a structural basis for the different ligand preferences of FPR1 and FPR2 (Supplementary Fig. 7A). To explore the molecular basis for the different receptor pharmacology, we generated a three-dimensional homology model of FPR1 based on the FPR2 structure and docked the same six formylpeptides to the FPR1 structural model. Two negatively charged residues at the top region of FPR2, E89 and D281[7.32], that form hydrogen bonding interactions with WKYMVm, are replaced by glycine residues in FPR1 (Supplementary Fig. 7B), which may explain the lower potency of WKYMVm for FPR1 than for FPR2[44]. Also, in the FPR1 structural model, there is a positively charged residue R84[2.63] at the similar position as E89 in FPR2, which together with the lack of negatively charged residues result in a positive charge distribution in the top region of FPR1 (Fig. 4c, d). This structural feature may lead to the high selectivity of FPR1 for short formylpeptides by providing a positive charge environment suitable for accommodating the C-terminal carboxyl group of short formylpeptides. Indeed, fMLF is about ~1000-fold more potent in activating FPR1 than activating FPR2[40]. Our docking results of fMLF in FPR1 also suggest that the last Phe residue in fMLF adopts a different conformation for binding to FPR1 compared to FPR2, whereas R84[2.63] in FPR1 forms a hydrogen bond with the carboxylate group of fMLF (Fig. 4d). This is consistent with the important role of R84[2.63] in fMLF binding to FPR1 demonstrated by previous mutagenesis studies[45,46]. Another residue in the ligand-binding pocket of FPR1, Y257[6.51], forms a hydrogen bond with the carbonyl group of N-formyl Met in fMLF, which is not conserved in FPR2 (Supplementary Fig. 7B), contributing to the high-affinity binding of fMLF to FPR1.

In the docking structures of FPR1 and FPR2, the N-terminal formyl group of formylpeptides is in a polar environment surrounded by residues D106[3.33], R201[5.38] and R205[5.42] in both receptors (Fig. 4b, d). This is similar to the acetamide group of the D-Met residue of WKYMVm (Fig. 2c). We propose that these three residues, which are highly conserved in FPR1 and FPR2 (Supplementary Fig. 7B), constitute a critical structural motif for recognizing the formyl group of formylpeptides as a pathogen-associated molecular pattern. It is also possible that the arrangement of D[3.33], R[5.38], and R[5.42] helps to keep the formyl group of formylpeptides in the appropriate conformation to allow the side chain of the formylated methionine residue to extend towards the 7TM core and activate the receptor. We further examined the effects of mutations D106[3.33]A, R201[5.38]A and

R205[5.42]A on formylpeptide fMLFII-induced FPR2 activation by GTPγS binding assays and bioluminescence resonance energy transfer (BRET) assays in HEK-293 cells (Fig. 4e). The results showed that R201[5.38]A and R205[5.42]A mutations could cause a dramatic decrease in fMLFII-induced FPR2 activation and G[i] coupling, similar to their effects in WKYMVm-induced FPR2 activation (Supplementary Fig. 4B), supporting the critical roles of R201[5.38] and R205[5.42] in the action of peptide ligands. For the D106[3.33]A mutation, we observed very little cell surface expression of the receptor in HEK-293 cells (Supplementary Fig. 4A). It is possible that this mutation compromises the overall structure of FPR2.

**Potential activation mechanism for FPR2**. There is no structure of inactive antagonist-bound FPR2 at the moment, although there are inactive structures of close homolog chemoattractant receptors C5aR, CRTH2, and BLT1[33,34,47]. Structure comparison with these inactive homolog receptors showed an outward displacement of TM6 and an inward shift of TM7 of FPR2 at the cytoplasmic region that are indicative of an active receptor conformation[39] (Fig. 5a), consistent with agonist-binding and G[i]-coupling. Among these receptors, C5aR is the closest phylogenetic neighbor of FPR2 as a peptide chemoattractant GPCR. Both C5aR and FPR2 have similar extracellular regions with overlapping peptide-binding pockets (Supplementary Fig. 8). The extracellular segments of TM6 in both receptors exhibit similar conformations as indicated by well aligned Q[6.52] residues (Fig. 5b). The outward shift of the cytoplasmic segment of TM6 in FPR2 compared to C5aR begins at position W[6.48] (Fig. 5b). W[6.48] and F[6.44] constitute a conserved structural motif in Class-A GPCRs[48,49]. Rearrangement of these two residues has been observed in a number of GPCRs that is associated with the displacement of the cytoplasmic region of TM6 and thus these two residues have been proposed to form a 'transmission switch' linking the extracellular agonist binding event to the conformational changes at the cytoplasmic region[48,49]. Therefore, it is likely that the rearrangement of the transmission switch in FPR2 links the ligand-binding event at the extracellular region to the conformational changes at the cytoplasmic region (Fig. 5b).

It is interesting to note that W254[6.48] in FPR2 is located at the narrow bottom region of the deep ligand-binding pocket (Fig. 2). As a result, the side chain of the D-Met residue in WKYMVm sits closely to W254[6.48] (Fig. 5b), which may induce conformational changes of this residue through steric effects, leading to further conformational changes of the transmission switch to activate the receptor. Given the similarity in the binding poses of formylpeptides and WKYMVm, it is likely that formylpeptides activate FPR2 through the same mechanism as WKYMVm. Also, considering the similar binding modes of formylpeptides for FPR1 as for FPR2 (Fig. 4), such a receptor activation mechanism may be conserved in FPR1 as well. Nevertheless, to gain more insights into the activation mechanism of FPR2, structures of inactive FPR2 are needed to do a more appropriate structural comparison analysis.

**Conserved and divergent features in FPR2 and G[i] coupling**. The structure of FPR2-G[i] complex revealed a similar mode of G[i] protein interactions as observed in the previously reported G[i]-coupled structures of rhodopsin, adenosine A1 receptor (A1AR), μOR and cannabinoid receptor 1 (CB1), and the structures of engineered G[o]-coupled 5-HT[1B] and G[oA]-coupled muscarinic M2 receptor[31,32,50–53]. In the structure of FPR2-G[i], the α-helical domain (AHD) of Gα[i] is disordered due to the absence of nucleotides. The N-terminal half of α5 (α5N) of Gα[i] inserts into the cavity at the cytoplasmic region of FPR2 (Fig. 6a).

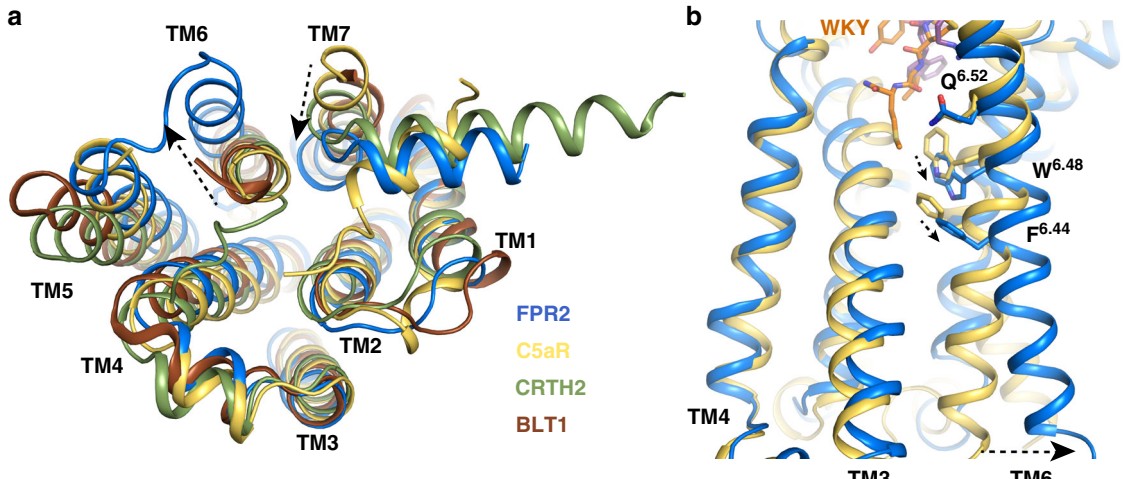

**Fig. 5 Potential FPR2 activation mechanism. a** Structural comparison of intracellular regions of FPR2 (blue) and other antagonist-bound chemoattractant GPCRs C5aR (yellow, PDB ID 6C1R), CRTH2 (green, PDB ID 6D26) and BLT1 (brown, PDB ID 5X33). **b** Structural comparison of TM6 of FPR2 and C5aR. In both figures, dashed arrows indicate potential conformational changes of FPR2 in activation based on structural comparison analysis.

Hydrophobic residues I344, L348, L353, and F354 on one side of α5N of Gα$_i$ cluster with surrounding hydrophobic residues from TM4, TM5, TM6, and ICL3 of FPR2 to form the major interaction interface. Polar residues on the other side of α5N of Gα$_i$ do not form direct contact with the receptor. The side chain of R123$^{3.50}$ in the conserved DR$^{3.50}$Y motif (DRC in FPR2) of the receptor extends towards TM7 on top of the α5 of Gα$_i$, forming a polar interaction network with the side chains of residues Y64 and Y221 of FPR2 and the carbonyl group of C351 of Gα$_i$. Such a polar interaction network is absent in other G$_i$-coupled structures of Class A GPCRs.

Despite the similarities in G$_i$ coupling to Class A GPCRs, the relative orientation of G$_i$ to the receptor is different in all structures, which is associated with notable differences in the G$_i$ and receptor interface. Compared to rhodopsin, A1AR, μOR and CB1, TM5, and TM6 in FPR2 are less extended (shorter) at the cytoplasmic region (Fig. 6b). As a result, the interface between FPR2 and the α5 of Gα$_i$ is the least extensive and the ICL3 in FPR2 is far away from the α4-β6 loop of Gα$_i$. No direct interactions are observed between FPR2 and the α4-β6 loop of Gα$_i$. In addition to the major site within the cytoplasmic cavity of FPR2 for interacting with Gα$_i$, FPR2 also interacts with G$_i$ at two other sites, the ICL2 and the helix 8. The ICL2 of FPR2 is close to the αN-β1 loop and the β2-β3 loop of Gα$_i$ (Fig. 6c). Residue V131 in the ICL2 is involved in a hydrophobic interaction network with residues L194, F196, and F336 from Gα$_i$. A similar pattern of interactions is observed in the structures of G$_i$-coupled A1AR and μOR. However, different from rhodopsin, A1AR, μOR and CB1, FPR2 engages in additional polar interactions through ICL2 with the αN-β1 loop and the β2-β3 loop of Gα$_i$ (Fig. 6c). The C-terminal end of helix 8 of FPR2 is close to the β subunit of G$_i$. The side chains of two residues in helix 8, R312 and H315, form direct polar interactions with the side chain of D312 from the Gβ subunit (Fig. 6d). Those interactions are absent in other G$_i$-coupled structures of Class A GPCRs.

The structural changes in G$_i$ are similar to those observed in other structures of GPCR and G$_i$ complexes. Compared to the GDP-bound G$_i$, the interaction of α5N of Gα$_i$ with FPR2 results in a large displacement of α5, which is translated to the conformational changes of the β6-α5 loop and the α1 helix, leading to the release of the guanine nucleotide[54] (Supplementary Fig. 6). In addition, the N-terminal end of the α1 helix of Gα$_i$ extends by one more helical turn into the P loop. This extra

helical turn occupies the space for the phosphate groups of guanine nucleotides when bound to G$_i$ (Fig. 7a). As a result, the binding of GDP or GTP is completely precluded in the FPR2-G$_i$ complex. A similar conformation of the α1 helix of Gα$_i$ is also observed in the rhodopsin- G$_i$ complex but not in the G$_i$-coupled A1AR, μOR and CB1 structures (Supplementary Fig. 9).

The cytoplasmic regions including ICL2 and ICL3 in FPR1 and FPR2 are highly conserved. All residues in FPR2 that are involved in the interactions with Gi including the two residues in helix 8, R312 and H315, that form direct interactions with Gβ are all conserved in FPR1 (Supplementary Fig. 7B). Therefore, it is highly likely that FPR1 couples to G$_i$ in the same way as FPR2.

## Discussion

Pattern recognition by pattern-recognition receptors is a critical step in immune surveillance. The innate immune system can discriminate between host and foreign cells by detecting pathogen-associated molecular patterns to protect the hosts from infectious threats. Host-derived damage-associated molecular patterns, on the other hand, allow immune systems to clear damaged cells and play roles in inflammation and tissue repair. Classic pattern-recognition receptor (PRR) families including the Toll-like receptors (TLRs), C-type lectin receptors (CLRs), NOD-like receptors (NLRs), RIG-I like receptors (RLRs) and the AIM2-like receptors (ALRs) have been well characterized to recognize molecular patterns such as lipoproteins, lipopolysaccharide (LPS), lipids and nucleotides[55]. Numerous structures of these receptors with diverse ligands have greatly advanced our understanding of how they recognize diverse molecular patterns[56]. FPRs were classified as pattern-recognition receptors in some reviews since they recognize formylpeptides as a pathogen-associated molecular pattern[2–4,57]. However, so far, no structures of FPRs have been reported, and the molecular mechanisms underlying the recognition of formylpeptides as important molecular patterns from pathogens and host damaged cells are not clear. In this paper, we report a cryo-EM structure of WKYMVm-bound FPR2-G$_i$ complex. The structure together with our computational modeling and docking studies identified critical structural details in FPRs for recognizing formylpeptides. In addition, our results revealed different structural features in the extracellular regions

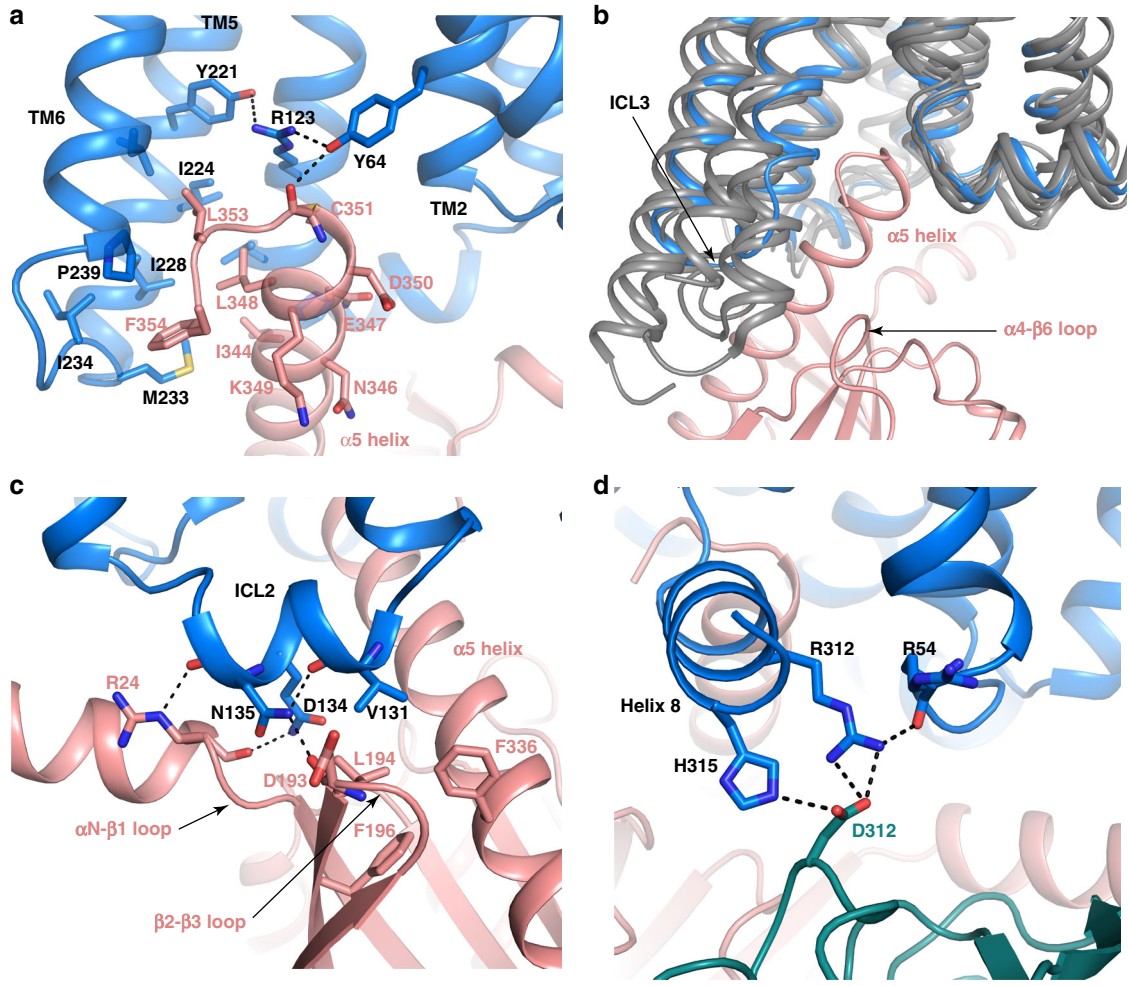

**Fig. 6 FPR2 and G_i interface. a** Interactions between α5N of Gα_i (salmon) and the receptor (blue) in the cavity at the cytoplasmic region of FPR2.
**b** Comparison of the cytoplasmic regions of TM5 and TM6 and ICL3 in FPR2 (blue) and in μOR (PDB ID 6DDE), A1AR (PDB ID 6D9H), CB1 (PDB ID 6N4B) and rhodopsin (PDB ID 6CMO) (all dark gray). Gα_i in the structure of FPR2-G_i is shown in salmon. **c** Interactions between the ICL2 of FPR2 and Gα_i.
**d** Interactions between FPR2 and Gβ.

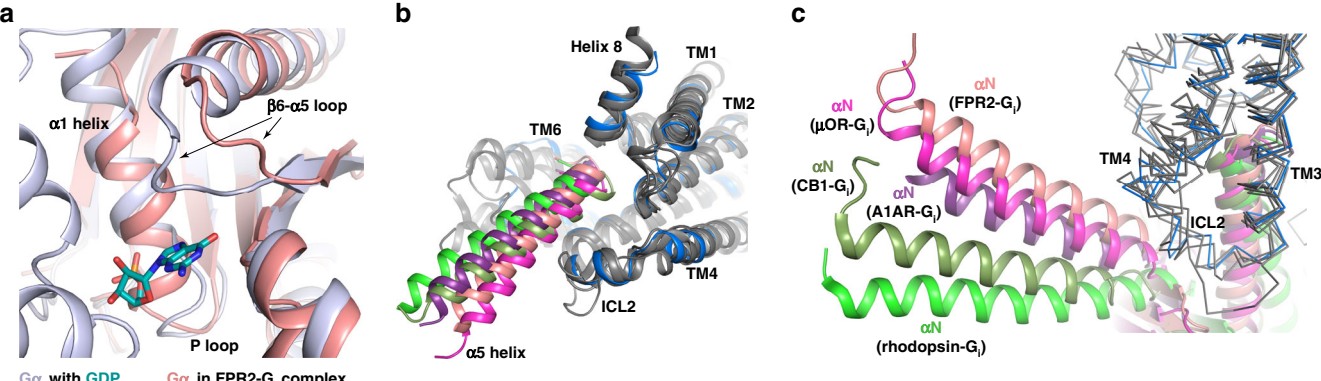

**Fig. 7 G_i protein in different structures. a** Structural comparison of Gα_i in the FPR2-G_i complex and GDP-bound Gα_i (PDB ID 1GP2). G_i in the FPR2-G_i complex is colored in salmon. G_i in the GDP-bound structure is colored in light blue. GDP is shown as green cyan sticks. **b, c** Structural comparison of α5 (**b**) and αN (**c**) of Gα_i in different structures. Gα_i is colored in salmon in FPR2-G_i, magenta in μOR-G_i, purple in A1AR-G_i, dark green in CB1-G_i and green in rhodopsin-G_i. The structural alignment is based on the receptors. FPR2 is colored in blue. All other receptors are colored in dark gray.

of FPR1 and FPR2 that lead to their different preferences of formylpeptides and thus different roles in host defense. Structural comparison analysis allowed us to define a potential activation mechanism for FPR2 by formylpeptides and WKYMVm.

Previous studies showed that besides recognition of formylpeptides, FPR2 plays far more complex roles in inflammation. A large number of structurally and functionally unrelated endogenous peptides act on FPR2 to promote inflammatory responses; several

other endogenous peptides or proteins and SPMs, on the other hand, can act on FPR2 to promote the resolution of inflammation[2,3,58]. It is speculated the FPR2 may respond to signaling molecules generated at different stages of inflammation to either promote or resolve inflammation[58]. Therefore, FPR2 may function as a checkpoint receptor in inflammation to maintain a balanced inflammatory process. Our structure revealed a remarkably open ligand-binding pocket with a vast space, which may allow FPR2 to recognize long peptides and large proteins such as annexin A1. The amphiphilic environment of the ligand-binding pocket is suitable for the binding of pro-resolving eicosanoid lipid molecules with amphiphilic structures. In addition, the MD simulation studies suggested a highly versatile nature of the extracellular region of FPR2 for recognizing chemically diverse ligands. The deep ligand-binding pocket may also allow an easy access to the transmission switch by diverse ligands to activate the receptor. This is in contrast to many other GPCRs, in which the transmission switch motif is much less accessible to ligands (Fig. 2d). Such a structural feature provides a possible molecular basis for why FPR2 can be activated by many chemically diverse ligands.

To date, several structures of $G_i$-coupled Class A GPCRs, including rhodopsin, A1AR, μOR, and CB1, and structures of engineered $G_o$-coupled 5HT1B and $G_{oA}$-coupled muscarinic M2 receptor have been reported. Compared to those structures, the structure of FPR2-$G_i$ exhibits an overall similar interaction pattern but noticeable differences. In fact, the orientation of the α5 of $Gα_i$, the major receptor interaction site on $G_i$, relative to the receptor is different among all structures (Fig. 7b). This is associated with different orientations and positions of the whole $G_i$ heterotrimer relative to the receptor. If we align all the structures based on the receptors, we can clearly see that the αN of $Gα_i$ also shows a large deviation. While the αN of $Gα_i$ in A1AR-$G_i$, μOR-$G_i$, FPR2-$G_i$ and 5HT1B-$G_o$ are close to each other, the αN of $Gα_i$ in CB1-$G_i$ is closer to it in the rhodopsin-$G_i$ (Fig. 7c). This may suggest at least two different profiles of $G_i$ coupling for Class A GPCRs. It is also possible that the nucleotide-free state of GPCR-G protein complexes is not a single state, and the structures of nucleotide-free GPCR-$G_i$ complexes may represent different conformation states in the dynamic G protein activation process. In fact, the conformation of the β6-α5 loop, which is important for nucleotide binding[54], is different among all $G_i$-coupled Class A GPCR structures (Supplementary Fig. 9). Very recently, a structure of $G_i$-coupled Smoothened, a Class F GPCR, was reported, which revealed a different arrangement of $G_i$ protein in the complex compared to that of all other class A GPCR-$G_i$ complexes[59]. All of those structural results suggest a high versatility of $G_i$ for coupling to GPCRs and different GPCRs may bind to and activate $G_i$ through different ways.

Although our structure provides mechanistic insights into the ligand recognition, receptor activation and $G_i$ protein coupling for the FPR family, the mechanism for the functional promiscuity of FPRs is still puzzling. More structures of FPR2 with other ligands, especially lipid and synthetic small-molecule agonists, are needed to further understand how FPR2 recognizes chemically diverse ligands. In addition, how FPR2 transmits signals from diverse ligands to the intracellular pace to play distinct, sometimes opposite, physiological roles in inflammation is largely unknown. A deep molecular understanding of FPR2 functional promiscuity is critical for the development of biased FPR2 ligands, which may hold the promise of becoming a new class of drugs that can actively promote the resolution of inflammation for the treatment of a broad spectra of inflammatory diseases.

## Methods

**Constructs design**. The coding sequence of wild type human FPR2 (residues 1–342) was cloned into pFastbac (ThermoFisher) with an N-terminal FLAG tag followed by a TEV cleavage site, and a C-terminal HIV 3C protease

site–oMBP–MBP–His8 tag to facilitate expression and purification. The prolactin precursor sequence was inserted into the N terminal to increase the protein expression. A dominant-negative bovine Gαi1 (DNGαi1) construct was generated by site-directed mutagenesis to incorporate mutations G203A and A326S to decrease the affinity of nucleotide binding and increase the stability of Gαβγ complex[29]. All the three G protein complex components, DNGαi1, rat Gβ1 and bovine Gγ2, were cloned into pFastbac individually. Primer sequences used for making constructs of FPR2, $G_i$ protein and scFv16 and introducing mutations in FPR2 are summarized in Supplementary Table 2.

**Preparation of scFv16**. Coding sequence of scFv16 was constructed into pFastbac vector (ThermoFisher) with a GP67 signaling peptide inserted into the N-terminal and a Tev cleavage-His8 at the C-terminal. The purification of scFv16 was conducted as previously described (Koehl et al.[31]). In brief, secreted scFv16 from Sf9 insect cell culture infected by baculovirus was purified using Ni-NTA and size exclusion chromatography. After balancing the pH and removing the chelating agents by $Ni^{2+}$ and $Ca^{2+}$, the cell culture supernatant was loaded into Ni-NTA. The Nickel resin was firstly washed with 20 mM HEPES pH 7.2, 100 mM NaCl, 50 mM imidazole for 10 column volumes and then eluted in buffer containing 250 mM imidazole. The eluted sample was treated with TEV protease (homemade) followed by dialysis in 20 mM HEPES pH 7.2, 100 mM NaCl overnight and then reloaded onto Ni-NTA resin to remove cleaved octa-histidine tag. The flow through was collected and applied to a HiLoad 16/600 Superdex 200 pg column (GE Healthcare) with buffer 20 mM HEPES pH 7.2, 100 mM NaCl. The monomeric peak fractions were concentrated and fast-frozen by liquid nitrogen.

**Protein complex expression and purification**. FPR2, DNGαi1, His8-tagged Gβ1 and Gγ2 were co-expressed in Sf9 insect cells (Novagen) using the Bac-to-Bac baculovirus expression system (ThermoFisher). Cell cultures were grown in ESF 921 serum-free medium (Expression Systems) to a density of $3.5 × 10^6$ cells/mL and then infected with the four types of baculovirus expressing FPR2, DNGαi1, His8-tagged Gβ1 and Gγ2 at the ratio of 1:1:1:1. 48 hours after infection, the cells were collected by centrifugation at $1000 × g$ (ThermoFisher, H12000) for 20 min and kept frozen at −80 °C for further usage.

For the purification of FPR2-$G_i$ complex, cell pellets from 2 L culture were thawed at room temperature and suspended in 20 mM HEPES pH 7.2, 50 mM NaCl, 5 mM $CaCl_2$, 5 mM $MgCl_2$. Complex was formed on membrane in the presence of 0.5 μM WKYMVm peptide (GL Biochem) and treated with apyrase (25 mU mL$^{-1}$, NEB), followed by incubation for 1.5 h at room temperature. Cell membranes were collected by ultra-centrifugation at 100,000 x g for 35 min. The membranes were then re-suspended and solubilized in buffer containing 20 mM HEPES, pH 7.2, 100 mM NaCl, 25 mM imidazole, 5 mM $CaCl_2$, 10% glycerol, 0.5% (w/v) lauryl maltose neopentylglycol (LMNG, Anatrace), 0.1% (w/v) cholesteryl hemisuccinate TRIS salt (CHS, Anatrace), 0.1%(w/v) digitonin (Sigma), 0.5 μM WKYMVm and 25 mU mL$^{-1}$ apyrase for 3 h at 4 °C. The supernatant was isolated by centrifugation at $100,000 × g$ for 45 min and then incubated overnight at 4 °C with pre-equilibrated Nickel-NTA resin. After batch binding, the nickel resin with immobilized protein complex was manually loaded onto a gravity column. The resin was washed with 10 column volumes of 20 mM HEPES, pH 7.2, 100 mM NaCl, 25 mM imidazole, 0.01% LMNG (w/v), 0.002% CHS (w/v), 0.1% digitonin (w/v), 0.5 μM WKYMVm and eluted with the same buffer plus 300 mM imidazole. The Ni-NTA eluate was further incubated by batch binding to 2 mL amylose resin (NEB) for 2 h at 4 °C. Detergent was exchanged on resin by a series of washing steps in 20 mM HEPES, pH 7.2, 100 mM NaCl, 0.5 μM WKYMVm supplemented with different detergents: first 0.01% LMNG, 0.002% CHS, 0.1% digitonin, then 0.002% LMNG, 0.0004% CHS, 0.1% digitonin, and finally 0.1% digitonin for 10 column volumes each. Subsequently, the amylose resin with bound material was treated with HIV 3C protease (homemade) and 1.8 mg scFv16 for 2 h at room temperature. Released protein was concentrated and then loaded onto a Superdex 200 10/300 GL increase column (GE Healthcare) pre-equilibrated with buffer containing 20 mM HEPES, pH 7.2, 100 mM NaCl, 0.075% digitonin, 0.5 μM WKYMVm. After a second column separation, the eluted fractions of monomeric complex were collected and concentrated for electron microscopy experiments. The final yield of purified complex is ~0.75 mg per liter of insect cell culture.

**Preparation of vitrified specimen**. For cryo-EM grid preparation, 2.5 μL purified FPR2-$G_i$-scFv16- WKYMVm complex at the concentration of 10.3 mg mL$^{-1}$ was applied to an EM grid (Quantifoil, 300 mesh Au R1.2/1.3, glow discharged for 1 min using a Harrick plasma cleaner (Harrick)) in a Vitrobot chamber (FEI Vitrobot Mark IV). Protein concentration was determined by absorbance at 280 nm using a Nanodrop 2000 Spectrophotometer (Thermo Fisher Scientific). The Vitrobot chamber was set to 95% humidity at 4 °C. The sample was blotted for 2 s before plunge-freezing into liquid ethane.

**Cryo-EM data acquisition**. Cryo-EM movie stacks were collected on a Titan Krios microscope operated at 300 kV under EFTEM mode. Nanoprobe with 1μm illumination area was used. Data were recorded on a post-GIF Gatan K2 summit camera at a nominal magnification of 130,000, using super-resolution counting model. Bioquantum energy filter was operated in the zero-energy-loss mode with an energy slit

width of 20 eV. Data collection were performed using SerialEM with one exposure per hole. The dose rate is ~8.4 e−/ Å²/s. The total accumulative electron dose is ~67 e−/Å² fractioned over 40 subframes with a total exposure time of 8 s. The target defocus range was set to −1.3 to −1.9 μm.

**Data processing and 3D reconstruction**. A total of 5387 movie stacks were collected. Each movie stack was aligned, dose weighted and binned by 2–1.029 Å per pixel using MotionCor2[60]. CTF was determined using CTFFIND4[61]. A total of 1,231,594 particles were auto-picked using a Laplacian-of-Gaussian filter and extracted in 256x256 pixels box using RELION 3.0[62]. All particles were subject to six rounds of reference-free 2D classification. Total 691,426 particles were selected to generated a 3D initial model and followed by 3D auto-refine. The 3D auto-refinement generated a density map of 3.56 Å resolution. A subset of 227,647 particles was selected after 3D classification, and the reconstruction was improved to 3.21 Å. Subsequent focused 3D classification was performed with a soft mask that only includes the FPR2 region of the map. A subset of 203,133 particles was selected for 3D refinement and CTF refinement. The global resolution of the final reconstruction is 3.17 Å. The resolution was estimated by applying a soft mask around protein density with the FSC 0.143 criteria. Local resolution map was calculated using RELION 3.0. Surface coloring of the density map was performed using UCSF Chimera[63].

**Model building and structure refinement**. The cryo-EM structure of μ-opioid receptor-Gᵢ Protein complex (PDB: 6DDE) was used as initial model for model rebuilding and refinement against the electron microscopy map. The model was docked into the electron microscopy density map using Chimera[63] followed by iterative manual adjustment and rebuilding in COOT[64]. Real space refinement and rosetta refinement were performed using Phenix programs [65]. The model statistics was validated using MolProbity[66]. Structural figures were prepared in Chimera and PyMOL (https://pymol.org/2/). The final refinement statistics are provided in Supplementary Table 1. The extent of any model overfitting during refinement was measured by refining the final model against one of the half-maps and by comparing the resulting map versus model FSC curves with the two half-maps and the full model.

**FPR1 modeling and computational docking**. Human FPR1 shares a very high sequence identity (68.7%) with FPR2. Considering this, the three-dimensional homology model of FPR1 was generated using the FPR2 structure as a template. Modeller[67] version 9.20 was used to generate 100 homology models from which the minimum DOPE[68] score model was selected for docking. FPR1 model was generated in complex with WKYMVm as in the structure of FPR2-Gᵢ complex.

The FPR2 structure and FPR1 homology model were prepared for docking using the protein-preparation wizard in Maestro (Schrödinger Release 2018-1: Maestro, Schrödinger). During the protein-preparation hydrogens were added and protonation states of titratable amino-acids were determined. Docking was then performed using GLIDE/SP-peptide[41] in Schrödinger. To ascertain the quality of the docking procedure, WKYMVm peptide was re-docked into the FPR2 structure and the results were checked to make sure the binding pose could be reproduced. After establishing the docking protocol, formylpeptides fMLF, fMLFII, fMLFIK, fMLFK, fMLFW, and fMLFE were docked in both FPR2 and FPR1. The docked poses were clustered based on the R.M.S.D (root mean squared deviation) and the best scoring poses with docked peptides in C-terminus in (C-ter-in) or N-terminus in (N-ter-in) conformation were extracted for further analysis. In the N-ter-in conformation, the N-formyl methionine of formylpeptides occupies the same space as D-Met in WKYMVm.

**System preparation and molecular dynamics simulation**. All-atom atmospheric molecular dynamics simulations of apo FPR2 transmembrane domain were performed using the CHARMM36m forcefield[69] with the GPU accelerated Particle-Mesh Ewald molecular dynamics (pmemd.cuda) engine within AMBER18 (https://ambermd.org/AmberMD.php). The receptor was prepared by removing all G protein subunits and all heteroatoms with exception of cholesterol molecules. The receptor was then aligned for membrane insertion with the Orientations of Proteins database PPM server[70] and inserted into a pre-equilibrated POPC lipid bilayer solvated in a box of TIP3P waters with 150 mM NaCl and neutralized by removing appropriate ions or counter ions using the Desmond system builder within Maestro (Schrödinger Release 2018-1: Maestro, Schrödinger). Titratable residues were left in their dominant state at pH 7.0 and all histidine side chains were represented with a hydrogen atom on the epsilon nitrogen. Free protein amino and carboxyl groups were capped with neutral acetyl and methylamine groups. Representative initial system dimensions were 85 × 85 × 100 Å and comprised of 134 lipids, 10,396 water molecule, 29 sodium ions and 40 chloride ions for a total of ~54,500 atoms.

Prior to production simulations, 25,000 steps of energy minimization were carried out followed by equilibration in the canonical NVT and isothermal-isobaric NPT ensembles for 10 and 50 ns, respectively with harmonic restraints (10 kcal mol⁻¹ Å⁻²) placed on all Cα atoms. Each system was then simulated for an additional 50 ns without harmonic restraints. Production simulations were performed with a 2 fs time-step in the NPT ensemble with semi-isotropic coupling at 310 K and 1 bar maintained by the Langevin thermostat and Monte Carlo barostat with periodic-boundary conditions. Bonds involving hydrogen atoms were constrained by SHAKE and with

non-bonded interactions cut at 8 Å. Trajectory snapshots were saved every 10 ps. System parameters and trajectories are available upon request.

**Thermal stability assay of protein complex**. To test the thermal stability of FPR2-Gᵢ-scFv16 complex, a fluorescence assay was conducted using the thiol-specific fluorochrome N-[4-(7-diethylamino-4-methyl-3-coumarinyl)-phenyl]-maleimide (CPM), which reacts with the free cysteines embedded in the protein. Prior to use, the CPM dye (Sigma Aldrich) was dissolved at 4 mg/mL in DMSO, and then diluted 1:40 with CPM dilution buffer containing 20 mM HEPES, pH 7.2, 100 mM NaCl. The protein (30–40 ng) was diluted in CPM dilution buffer supplemented with 0.1% digitonin to a final volume of 150 μL. 10 μL of the diluted dye was added and mixed together with protein sample. After incubation in ice for 10 min, the mixture was transferred into a sub-micro quartz fluorimeter cuvette (Starna Cells, Inc.). The melting curve was recorded by heating the mixture from 20 °C to 90 °C with a rate of 2 °C/min in a Cary Eclipse Fluorescence Spectrophotometer (Agilent Technologies). The excitation wavelength was set at 387 nm and the emission wavelength was 463 nm.

**³⁵S-GTPγS binding assays**. To form receptor-Gi complex, ~250 μg/ml of membrane of Sf9 cells overexpressing FPR2 or C5aR (for Fig. 1a) or membrane of HEK-293 cells transiently expressing FPR2 and mutants was incubated with 200 nM purified Gi protein for 25 minutes at 4°C in buffer containing 20 mM HEPES pH 7.5, 100 mM NaCl, 5 mM MgCl₂, 5 μM GDP, 0.1 μM TCEP, and 3 μg/ml BSA. Next, 25 μL aliquots of the pre-formed complex were mixed with 225 μL reaction buffer containing 20 mM HEPES, pH 7.5, 100 mM NaCl, 5 mM MgCl₂, 1 μM GDP, 0.1 μM TCEP, 3 μg/ml BSA, 35 pM ³⁵S-GTPγS (Perkin Elmer) and various concentrations of WKYMVm (GL Biochem). After 10–15 min reaction at 25 °C, 4 ml of cold wash buffer containing 20 mM HEPES pH 7.5, 100 mM NaCl and 5 mM MgCl₂ was added to terminate the reaction, and the membranes was collected by filtering through glass fiber prefilters (Millipore Sigma). After four times additional wash, the filters were incubated with 5 ml of CytoScint liquid scintillation cocktail (MP Biomedicals) and counted in a Beckman LS6500 scintillation counter to determine the receptor activation induced binding of ³⁵S-GTPγS to Gᵢ. The data were analyzed by GraphPad Prism 6 (GraphPad Software). Results are presented as mean ± s.e.m. from three independent experiments.

Receptor cell surface expression was determined by measuring the binding of Alexa-488 labeled anti-FLAG M1 antibody (homemade) to the cell surface by FACS.

**BRET assays**. HEK-293 cells (ATCC, Manassas, VA, USA) were propagated in plastic flasks and on 6-well plates according to the supplier's protocol. To measure Gᵢ coupling to FPR2, cells were transiently transfected with an FPR2-Rluc8 receptor (wt or mutant) and Gαᵢ₁, Venus-1-155-Gγ₂ and Venus-155-239-Gβ₁ using linear poly-ethyleneimine MAX (PEI MAX; MW 40,000) at an N/P ratio of 20 and were used for experiments 48 h later. Up to 3.0 μg of plasmid DNA was transfected in each well of a 6-well plate. To measure BRET signals, cells were washed twice with permeabilization buffer (KPS) containing 140 mM KCl, 10 mM NaCl, 1 mM MgCl₂, 0.1 mM KEGTA, 20 mM NaHEPES (pH 7.2), harvested by trituration, permeabilized in KPS buffer containing 10 μg ml⁻¹ high purity digitonin, and transferred to opaque black 96-well plates containing diluted ligands and 2U ml⁻¹ apyrase. After addition of coelenter-azine h (5 μM; Nanolight, Pinetop, AZ, USA), steady-state BRET measurements were made using a Mithras LB940 photon-counting plate reader (Berthold Technologies GmbH, Bad Wildbad, Germany). Raw BRET signals were calculated as the emission intensity at 520–545 nm divided by the emission intensity at 475–495 nm. The data were analyzed in Prism (GraphPad) by using 'log(agonist) vs normalized response-variable slope (four parameters)' of dose-response stimulation.

**Reporting summary**. Further information on research design is available in the Nature Research Reporting Summary linked to this article.

## Data availability

The 3D cryo-EM density maps have been deposited in the Electron Microscopy Data Bank under the accession numbers EMD-20126. Atomic coordinates for the atomic model have been deposited in the Protein Data Bank under the accession numbers 6OMM. The source data underlying Figs. 1a, 4e and Supplementary Figs. 4A and B are provided as a Source Data file. Other data are available from the corresponding authors upon reasonable request.

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

## Acknowledgements

The data were collected at the David Van Andel Advanced Cryo-EM facility at the Van Andel Research Institute. This work was supported by the National Institutes of Health grants 1R35GM128641 (to C.Z.), 1R01GM127710 (to H.E.X) and 1R01GM130142 (to N.L.), the Van Andel Research Institute funding (to K.M. and H.E.X.), XDB08020303 (to H.E.X.), and funding support from the Biomedical Research Council of A*STAR including the Industry Alignment Fund Pre-Positioning (H18/01/a0/C14) (to H.F. and R.K.V.). Y.Z. acknowledges the support from the UCAS joint Ph.D. Training Program.

## Author contributions

Y.Z., Y.K., and H.L. designed the expression constructs and purified the FPR2-Gi complex. Y.Z. prepared the final samples for cryo-EM data collection toward the structures, performed screening of cryo-EM grids and data collection and processing. H.L. performed the GTPγS binding assays and prepared constructs of FPR2 mutants. W.J. performed the BRET assays. E.Z. built and refined the structure models. T.X. assisted cryo-EM structure determination. P.W. performed MD simulations. R.K.V. built FPR1 model and performed docking studies under the supervision of H.F. X.M. and G.Z. assisted in cryo-EM data collection and processing. L.W. assisted cloning and protein expression. H.E.X., K.M., N.L., and C.Z. supervised all studies, and C.Z. wrote the paper with H.E.X., H.F., R.K.V., K.M., Y.Z., H.L., E.Z., and P.W.

## Competing interests

The authors declare no competing interests.
