## [Peer Review File · Nature Communications]

Reviewers' comments:

Reviewer #4 (Remarks to the Author):

This manuscript describes the structure of the FPR2-Gi complex bound to a peptide agonist by cryo-EM. Notably, this is a novel structure of a class A GPCR that has not previously been solved by crystallography, and therefore of interest to the GPCR structural biology community in itself. The structure shows an unusually wide-open conformation of the ligand binding site, and displays features that are discussed in light of the ligand binding promiscuity of this receptor. The study concludes with rationalizing structure-activity relationships of ligands by molecular docking, and a model of the receptor inactive state by molecular dynamics simulation.

This structure is a fine technical achievement, however, the authors fall short of convincingly validating it experimentally i.e. by mutagenesis. While it is true that there is previously published mutagenesis data that is also cited by the authors, the authors would do well to provide their own functional and/or binding data that 1) reproduces at least some of the most important literature data, and 2) probe novel features of this receptor experimentally and not only by simulation. Including at least some experimental controls carried out in the laboratory of the authors would lend credibility to their other conclusions. It might also be helpful to the reader to compile the cited functional/affinity data numbers into a (supplementary) table rather than just providing references. As is, the amount of functional/binding data the authors provide compares unfavourably to other recent GPCR structural papers in similar journals.

The authors present two additional data points from BRET assays of two arginine binding site mutants in the revised version of their manuscript. From the associated curves it seems to me that the experiment does not reach saturating conditions – why was the concentration range not chosen such that the BRET intensity reaches a plateau? If unspecific binding becomes limiting at higher ligand concentrations it might be advantageous to choose a more sensitive assay, such as GTP γ S, which would have the additional advantage to make it better comparable to their figure 1. As it is the lower BRET values that are seen for arginine mutants could just result from lower expression. It would also be helpful to mark the location of these arginine residues in their structure (figure 2) and not only the docked complex of figure 4. How certain are the authors of the side-chain orientation of these residues? Without seeing the experimental density maps it is not possible to get a feeling for map quality and how well side chains are resolved, in particular those entertaining the ligand – a detailed view of the ligand in the binding site with electron density could be more valuable than a single number for global resolution.

Finally, the authors should clearly state how reproducible their molecular dynamics results are, and how many independent simulations were run. Since emphasis is put on the modeled inactive state of the receptor, the authors should show a more quantitative analysis of the MD simulation that goes beyond the current figure 5 that only shows a snapshot of before-after. It would be important to show time-evolution of RMSD as well as certain distances characteristic of the overall motion that the authors describe, perhaps in the supplement. This is important since the overall simulation time of 2 μ s, while perhaps state-of-the-art for all-atom simulations, seems very short for the type of motion to occur that the authors describe, i.e. the large-scale rearrangement of the TM bundle during receptor inactivation. The authors should explicitly discuss how their microsecond simulation/conformational change fits with the timescale of (in)activation of GPCRs and bacteriorhodopsin for which time-resolved experimental structural data is now accumulating (e.g. 10.1126/science.aaw8634 and 10.1073/pnas.1900261116). It will be vital to exclude simulation artifacts – in particular, the structure has been solved in detergent and in complex with ligand and G protein, but the simulation is carried out in a more biologically relevant lipid environment; embedding the system in lipid alone could conceivably cause the observed compaction of the binding site, in particular on the short timescale (hundreds nanoseconds) that is

covered by the MD simulation. It is impossible to judge if this results from insufficient equilibration of the system if no graphs are shown that quantify the conformational evolution of the system. To this end, it would also be prudent to carry out an independent simulation of the ligand-bound receptor in lipid environment - a compaction similar to that observed for the apo-state could be indicative of a detergent-induced artificial conformation of their initial experimental structure.

Manuscript ID: NCOMMS-19-28664A

Title: Structure of formylpeptide receptor 2-G_i complex reveals novel insights into ligand recognition and signaling

We thank the reviewer for the constructive comments. Please see our detailed responses to the comments below. The reviewer's comments are in blue font and our responses are in black font.

Reviewers' comments:

Reviewer #4 (Remarks to the Author):

This manuscript describes the structure of the FPR2-G_i complex bound to a peptide agonist by cryo-EM. Notably, this is a novel structure of a class A GPCR that has not previously been solved by crystallography, and therefore of interest to the GPCR structural biology community in itself. The structure shows an unusually wide-open conformation of the ligand binding site, and displays features that are discussed in light of the ligand binding promiscuity of this receptor. The study concludes with rationalizing structure-activity relationships of ligands by molecular docking, and a model of the receptor inactive state by molecular dynamics simulation.

This structure is a fine technical achievement, however, the authors fall short of convincingly validating it experimentally i.e. by mutagenesis. While it is true that there is previously published mutagenesis data that is also cited by the authors, the authors would do well to provide their own functional and/or binding data that 1) reproduces at least some of the most important literature data, and 2) probe novel features of this receptor experimentally and not only by simulation. Including at least some experimental controls carried out in the laboratory of the authors would lend credibility to their other conclusions. It might also be helpful to the reader to compile the cited functional/affinity data numbers into a (supplementary) table rather than just providing references. As is, the amount of functional/binding data the authors provide compares unfavourably to other recent GPCR structural papers in similar journals.

We thank the reviewer for the suggestions. Now we have included data on 8 mutations of FPR2 residues that interact with WKYMVm in Supplementary Figure 4, which is also shown below. We cited this data in the main content to support our structural findings. The data clearly showed that two arginine residues, R201^{5,38} and R205^{5,42}, at the bottom of ligand-binding pocket play important roles in the action of WKYMVm on FPR2. These two residues have been proposed to constitute a critical motif for recognizing formylpeptides. We believe that the amount of functional data we now included in our revised manuscript is comparable to or more than the amount of functional data in the recent papers of structures of GPCR-G protein complexes published in Nature, Science or Cell (PMIDs 29899450, 29899455, 29925951, 29925945, 30639101, 31073061).

As suggested by the reviewer, we now included a summary of cited functional data on the six formylpeptides we docked to our structure in Supplementary Table 2.

Supplementary Figure 4A and B

The authors present two additional data points from BRET assays of two arginine binding site mutants in the revised version of their manuscript. From the associated curves it seems to me that the experiment does not reach saturating conditions – why was the concentration range not chosen such that the BRET intensity reaches a plateau? If unspecific binding becomes limiting at higher ligand concentrations it might be advantageous to choose a more sensitive assay, such as GTP γ S, which would have the additional advantage to make it better comparable to their figure 1. As it is the lower BRET values that are seen for arginine mutants could just result from lower expression.

To determine ligand-induced receptor activation, we have performed the direct recruitment BRET assays, which monitor directly the recruitment of G proteins to GPCRs as a result of receptor activation in a format where the receptor is fused to a luciferase (Rluc8) and the G protein is fused to a BRET acceptor. The readout of this assay is ratiometric and depends on the FRACTION of GPCR-Rluc8 fusions that are in close proximity to G proteins at a given time. This relies on the local availability of G protein and the efficiency of G protein recruitment, but not on the absolute number of BRET donor GPCR-Rluc8 fusions present (provided G protein is in excess). In this regard, lower expression of GPCRs does not necessarily result in lower BRET values. By comparison, most cell-based signaling assays (e.g. cAMP accumulation measurement) depend on the absolute density of receptor expression at the cell surface, which requires careful adjustment when comparing receptors (and mutants) with variable trafficking efficiency.

Nevertheless, to provide more functional data to support the important roles of R201^{5.38} and R205^{5.42} in ligand recognition, as suggested by the reviewer, we performed GTP γ S binding assays on wtFPR2 and R201^{5.38}A and R205^{5.42}A mutants. We have adjusted their expression to make sure their expression levels on the cell surface were comparable (Supplementary Figure 4). The results as shown below are consistent with BRET assays. We now included data from both GTP γ S binding assays and BRET assays in Figure 4E in the revised manuscript.

Regarding the concentration range of ligand, since the ligand is a peptide with a high hydrophobicity (fMLFII), concentrations above 10 μ M could lead to fractions of aggregation. The concentration range used in our assays (1nM to 10 μ M for GTP γ S binding assays and 1pM to

10 μ M for BRET assays) was sufficient to show differences in the action of fMLFII on wtFPR2 and the mutants.

Figure 4E

It would also be helpful to mark the location of these arginine residues in their structure (figure 2) and not only the docked complex of figure 4. How certain are the authors of the side-chain orientation of these residues? Without seeing the experimental density maps it is not possible to get a feeling for map quality and how well side chains are resolved, in particular those entertaining the ligand – a detailed view of the ligand in the binding site with electron density could be more valuable than a single number for global resolution.

We thank the reviewer for the suggestion. We have now shown cryo-EM map for major regions of the structure including WKYMVm ligand in Supplementary Figure 3. We have also included the map for residues D106, R201 and R205 in Supplementary Figure 4C as shown below. The cryo-EM map for these residues and the ligand is clear. We revised Figure 2C to clearly show the location of D106, R201 and R205 and the ligand.

Finally, the authors should clearly state how reproducible their molecular dynamics results are, and how many independent simulations were run. Since emphasis is put on the modeled inactive state of the receptor, the authors should show a more quantitative analysis of the MD simulation that goes beyond the current figure 5 that only shows a snapshot of before-after. It would be important to show time-evolution of RMSD as well as certain distances characteristic of the overall motion that the authors describe, perhaps in the supplement. This is important since the overall simulation time of 2 μ s, while perhaps state-of-the-art for all-atom simulations, seems very short for the type of motion to occur that the authors describe, i.e. the large-scale rearrangement of the TM bundle during receptor inactivation. The authors should explicitly discuss how their microsecond simulation/conformational change fits with the timescale of (in)activation of GPCRs and bacteriorhodopsin for which time-resolved experimental structural data is now accumulating (e.g. 10.1126/science.aaw8634 and 10.1073/pnas.1900261116). It will be vital to exclude simulation artifacts – in particular, the structure has been solved in detergent and in complex with ligand and G protein, but the simulation is carried out in a more biologically relevant lipid environment; embedding the system in lipid alone could conceivably cause the observed compaction of the binding site, in particular on the short timescale (hundreds nanoseconds) that is covered by the MD simulation. It is impossible to judge if this results from insufficient equilibration of the system if no graphs are shown that quantify the conformational

evolution of the system. To this end, it would also be prudent to carry out an independent simulation of the ligand-bound receptor in lipid environment - a compaction similar to that observed for the apo-state could be indicative of a detergent-induced artificial conformation of their initial experimental structure.

We thank the reviewer for bringing up this important question. It has been previously shown that another GPCR β 2-adrenergic receptor could transition from the active state to the inactive state if its agonist and active conformation-stabilizing protein partner were removed (PMID 21228876). We speculated that apo-FPR2 without agonist and G_i protein might shift towards the inactive state during MD simulation similarly to β 2-adrenergic receptor. However, we agree with the reviewer that a 2- μ s timescale may not be long enough for the inactivation process to occur. Therefore, we have revised our discussion of FPR2 activation in the section 'Potential activation mechanism for FPR2' by removing the structural comparison of active FPR2 to simulated apo-FPR2. Instead, we now compared the active FPR2 structure to inactive structures of other closely related chemoattractant GPCRs and proposed a potential activation mechanism for FPR2 in this section. We also stated at the end of the section that " Nevertheless, to gain more insights into the activation mechanism of FPR2, structures of inactive FPR2 are needed to do a more appropriate structural comparison analysis".

We added one more paragraph at the end of section 'A widely open ligand-binding pocket for WKYMVm' to describe our MD simulation results on apo-FPR2. As suggested by the reviewer, we have shown time-evolution of RMSD and distances between the cytoplasmic ends of TM6 and TM2 and between ECL2 and ECL3 in Supplementary Figure 5. We stated that "In both simulations, the cytoplasmic end of TM6 moved towards TM2, which suggests a relaxation of TM6 towards the inactive conformational state likely due to the removal of agonist and G_i protein". We also discussed large conformational fluctuations of ECL2 and ECL3 in simulations. We acknowledged that MD simulations at 2 μ s timescale may not be enough to probe real conformational changes of the receptor.